# Beyond Single Views: Achieving Significant Gains in Text Clustering via Informative Diversification

## Abstract

Clustering text into coherent groups is a long-standing challenge, complicated by high-dimensional embeddings, semantic ambiguity, and distributional shifts in unseen data. Recent advances in large language models (LLMs) and retrieval-augmented generation (RAG) systems have further underscored the need for robust and scalable knowledge representation methods. In this work, we introduce a novel clustering framework based on informative diversification. Our method applies a set of semantic-preserving transformations to generate multiple views of the data, and then harnesses their collective structure through a spectral consensus process. We prove that consensus clustering achieves an exponentially lower expected error rate compared to any single view, provided the views are diverse and informative. We then propose an iterative co-training procedure that learns a cluster-friendly latent space by jointly minimizing a contrastive InfoNCE loss and a Gaussian mixture negative log-likelihood loss. This training sharpens assignments and pulls embeddings toward their cluster centroids, while dynamically updating cluster assignments to accommodate the evolving latent space. The result is a robust and generalizable model that not only outperforms baselines on benchmark datasets but also maintains strong accuracy on unseen text, making it a powerful tool for real-world knowledge discovery and retrieval-augmented generation systems.

***Keywords:*** *Informative Diversification, Consensus Clustering, Multi-View Embeddings, Gaussian Mixture Models, Contrastive Learning.*

## 1 Introduction and Previous Work

The unprecedented growth of unstructured text—ranging from scientific repositories and enterprise communications to social media and multilingual streams—has made the discovery of latent thematic structures indispensable. Since its inception in the 1960s for organizing bibliographic records, document clustering has evolved through methods such as $k$-means, hierarchical clustering, and probabilistic models including Latent Dirichlet Allocation (LDA) (Blei et al. (2003)) and the Stochastic Block Model (SBM). Today, the role of clustering is further amplified by large language models (LLMs) and retrieval-augmented generation (RAG), where well-structured corpora are essential for curating training data, reducing redundancy, and strengthening retrieval pipelines. Despite decades of progress, clustering text remains difficult. Text embeddings are high-dimensional and sparse, semantics are context-dependent, and multilingual corpora complicate alignment.

Traditional text clustering methods have been dominated by three main families of algorithms: centroid-based approaches (K-Means), probabilistic models (Gaussian Mixture Model, Stochastic Block Model), and graph-based methods (Spectral Clustering, Modularity Maximization). A considerable amount of work has used probabilistic models as an effectively proven method for text clustering considering the high dimensional space representation of textual embeddings and the probabilistic nature of this task.

A foundational work for clustering with probabilistic models is the Expectation-Maximization algorithm for Gaussian Mixture Models (GMMs) established by (Dempster et al. (1977)). This probabilistic framework was successfully applied to text with generative models like Probabilistic Latent

Semantic Analysis (Hofmann (1999)) and Latent Dirichlet Allocation (Blei et al. (2003)), which were used to discover thematic structures. Subsequent developments like the Correlated Topic Model (Blei & Lafferty (2007)) extended these concepts to capture topic correlations. A significant drawback of these early approaches was their operation on simplistic bag-of-words representations, which ignored word order and contextual semantics, limiting their capacity to capture nuanced meaning.

The advent of deep learning catalyzed a shift towards jointly learning representations and cluster assignments. Early neural approaches like the Deep Clustering Network (Yang et al. (2016)) demonstrated how neural architectures could learn clustering-friendly representations. Other works like Deep Embedded Clustering (Xie et al. (2016)) integrated autoencoders with clustering objectives. ClusterGAN (Mukherjee et al. (2019)) employed generative adversarial networks to learn latent spaces amenable to clustering. While innovative, these methods exhibited a strong dependence on careful pre-training and initialization and were often trapped in suboptimal local minima, and relied on similarity measures ill-suited for complex, high-dimensional embeddings. This led to a resurgence of probabilistic thinking within deep architectures. For instance, Variational Deep Embedding (Jiang et al. (2017)) unified variational autoencoders with a GMM prior. This was extended by subsequent works like GraphEDM (Wang et al. (2019)) which incorporated graph convolutional networks to capture structural relationships. Although elegant, such deep generative models introduced considerable training complexity and instability by simultaneously optimizing reconstruction and clustering.

To combat the instability and variance inherent in single-model approaches, the field shifted towards learning multi-view and consensus clustering. Early consensus methods (Strehl & Ghosh (2002)) aggregated multiple clusterings into a robust partition but were computationally expensive and operated as post-hoc procedures disconnected from representation learning. Recent innovations in self-supervised consensus learning (Liu et al. (2021)) have attempted to generate synthetic views through semantic-preserving transformations, reducing the dependency on naturally occurring multiview data while maintaining the stability benefits of consensus approaches.

**Contributions:** In our work, we build on this progression with the following contributions:

1. We propose a clustering method that creates multiple views of the original embeddings and then harnesses their collective structure through a spectral consensus process, reducing misclustering error relative to single-view methods.

2. We design a hybrid objective combining contrastive learning and gaussian mixture negative log-likelihood, which maximizes mutual information between embeddings and consensus clusters, improving generalization to unseen data.

3. We develop an iterative co-training scheme that alternates between updating cluster assignments and model parameters, yielding stable solutions that outperform baselines.

Figure 1 illustrates the methodology including: Consensus clustering and Contrastive Training.

## 2 METHODOLOGY

### 2.1 OVERVIEW OF THE GAUSSIAN MIXTURE MODEL (GMM)

The Gaussian Mixture Model (GMM) is a probabilistic model for clustering and density estimation. It is defined as a weighted sum of Gaussian distributions:

$$p(\mathbf{x}) = \sum_{k=1}^{K} \pi_k \mathcal{N}(\mathbf{x} \mid \mu_k, \Sigma_k),$$

where:

- $\pi_k$ is the mixing coefficient of the $k$-th component, with $\sum_{k=1}^{K} \pi_k = 1$,
- $\mathcal{N}(\mathbf{x} \mid \mu_k, \Sigma_k)$ denotes the Gaussian distribution with mean $\mu_k$ and covariance $\Sigma_k$.

Given a dataset $\mathcal{D} = \{\mathbf{x}_1, \dots, \mathbf{x}_N\}$, the likelihood under the GMM is: $\mathcal{L}(\Theta) = \prod_{n=1}^{N} p(\mathbf{x}_n \mid \Theta)$, where $\Theta = \{\pi_k, \mu_k, \Sigma_k\}_{k=1}^{K}$ are the model parameters to be estimated.

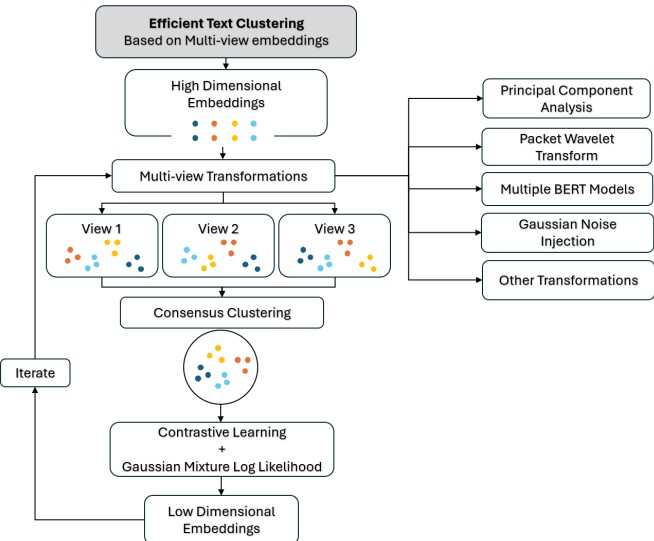

Figure 1: *Text Clustering using Multi-view Transformation*

The Expectation-Maximization (EM) algorithm is used for maximum likelihood estimation, until convergence:

- **E-Step:** Compute the responsibilities:

$$\gamma_{nk} = \frac{\pi_k \mathcal{N}(\mathbf{x}_n \mid \mu_k, \Sigma_k)}{\sum_{j=1}^{K} \pi_j \mathcal{N}(\mathbf{x}_n \mid \mu_j, \Sigma_j)}.$$

- **M-Step:** Update parameters:

$$\pi_k \leftarrow \frac{1}{N} \sum_{n=1}^{N} \gamma_{nk}, \quad \mu_k \leftarrow \frac{\sum_{n=1}^{N} \gamma_{nk}\mathbf{x}_n}{\sum_{n=1}^{N} \gamma_{nk}}, \quad \Sigma_k \leftarrow \frac{\sum_{n=1}^{N} \gamma_{nk}(\mathbf{x}_n - \mu_k)(\mathbf{x}_n - \mu_k)^T}{\sum_{n=1}^{N} \gamma_{nk}}.$$

## 2.2 TEXT CONSENSUS CLUSTERING BASED ON MULTI VIEW REPRESENTATION

### 2.2.1 MULTI-VIEW EMBEDDINGS VIA VIEW-SPECIFIC TRANSFORMATIONS

Let $\boldsymbol{x}_i$ denote the textual content associated with the $i$-th document of a textual dataset where $i \in \{1, \ldots, N\}$. Each text is encoded with a Sentence-BERT Transformer $f_\theta$ to obtain the textual embeddings: $\boldsymbol{h}_i = f_\theta(\boldsymbol{x}_i), \quad \boldsymbol{h}_i \in \mathbb{R}^d$.

Given embeddings $\mathbf{H} = [\mathbf{h}_1, \ldots, \mathbf{h}_n] \in \mathbb{R}^{d \times n}$, we construct $m$ alternative views using view-specific transformations. For transformation $T_\theta : \mathbb{R}^d \to \mathbb{R}^d$,

$$\mathbf{H}_i^{(v)} = T_{\theta^{(v)}}(\mathbf{h}_i), \quad v \in \{1, \ldots, m\}, \ i \in \{1, \ldots, n\},$$

where each $\theta^{(v)}$ is a randomly sampled parameter set (e.g., a random matrix or noise parameter) for view $v$. This results in $m$ distinct transformed datasets $\mathbf{H}^{(v)} = [\mathbf{h}_1^{(v)}, \ldots, \mathbf{h}_n^{(v)}]$. We consider both deterministic transformations (such as using Principle Component Analysis PCA, Wavelet Packet Transforms WPT, or the use of several BERT models as shown in Table 1) and stochastic transformations (such as injecting Gaussian Noise).

### 2.2.2 CONSENSUS CLUSTERING VIA SPECTRAL CLUSTERING

For each view $v$, we perform clustering using GMM with $K$ components and isotropic homogeneous covariance. Let $\mathcal{C}^{(v)} = \{C_1^{(v)}, \ldots, C_K^{(v)}\}$ denote the resulting cluster assignment for view $v$, and $\mathcal{A}^{(v)} : \{1, \ldots, n\} \to \{1, \ldots, K\}$ the cluster assignment function: $\mathcal{A}^{(v)}(i) = k$ if $\mathbf{x}_i^{(v)} \in C_k^{(v)}$.

Table 1: Several Pretrained sentence embedding models used for the Multi-view embeddings generation.

| Model | Output dimension |
|---|---|
| all-MiniLM-L6-v2 | 384 |
| paraphrase-MiniLM-L6-v2 | 384 |
| multi-qa-MiniLM-L6-cos-v1 | 384 |
| all-mpnet-base-v2 | 768 |
| paraphrase-multilingual-MiniLM-L12-v2 | 384 |
| distiluse-base-multilingual-cased-v2 | 512 |
| all-distilroberta-v1 | 768 |

We construct a co-occurrence (consensus) matrix $\mathbf{W} \in \mathbb{R}^{n \times n}$, whose entry $\mathbf{W}_{ij}$ is the fraction of views in which nodes $i$ and $j$ are assigned to the same cluster:

$$\mathbf{W}_{ij} = \frac{1}{m} \sum_{v=1}^{m} \mathbb{I}\left(\mathcal{A}^{(v)}(i) = \mathcal{A}^{(v)}(j)\right), \text{ where } \mathbb{I}(\cdot) \text{ is the indicator function.}$$

Finally, we obtain the refined consensus clustering $\hat{\mathcal{C}} = \{\hat{C}_1, \ldots, \hat{C}_K\}$ with spectral clustering. We compute the normalized graph Laplacian $\mathbf{L}$ corresponding to $\mathbf{W}$:

$$\mathbf{D} = \text{diag}(d_1, \ldots, d_n), \quad d_i = \sum_{j=1}^{n} \mathbf{W}_{ij}, \quad \mathbf{L} = \mathbf{I} - \mathbf{D}^{-1/2} \mathbf{W} \mathbf{D}^{-1/2},$$

where $\mathbf{D}$ is the diagonal degree matrix and $\mathbf{I}$ is the identity matrix.

Then, we compute the $K$ smallest eigenvectors of $\mathbf{L}$ and form the matrix $U = [\mathbf{u}_1, \mathbf{u}_2, \ldots, \mathbf{u}_K] \in \mathbb{R}^{n \times K}$, where $\mathbf{u}_k$ denotes the $k^{\text{th}}$ eigenvector associated with the $k^{\text{th}}$ smallest eigenvalue.

We then row-normalize $U$ to obtain $\tilde{U}_i = \frac{U_i}{\|U_i\|_2}, \quad i = 1, \ldots, n$, where $U_i$ is the $i^{\text{th}}$ row of $U$.

Finally, we apply $K$-means clustering to the rows $\{\tilde{U}_1, \ldots, \tilde{U}_n\}$, assigning each node $i$ to its corresponding cluster $\hat{C}_k$ (refer to Algorithm 1 for implementation details).

### 2.2.3 Consensus vs. Single-View Clustering:

In the appendix, we provided theoretical guarantees showing that consensus clustering based on multiple transformed views of the data achieves a strictly lower expected misclustering rate compared to applying Gaussian Mixture Models (GMMs) on a single view. The purpose of this analysis was to rigorously quantify the advantage of aggregating information across multiple views rather than relying on any one view alone.

Specifically, we studied the minimax risk: $\inf_{\hat{z}} \sup_{z^*} \mathbb{E}[h(\hat{z}, z^*)]$,

The misclustering fraction is defined as:

$$h(\hat{z}, z^*) = \min_{\pi \in S_K} \frac{1}{n} \sum_{j=1}^{n} \mathbf{1}\{\pi(\hat{z}_j) \neq z_j^*\}.$$

where $z^* = (z_1^*, \ldots, z_n^*)$ denotes the true cluster assignment of all $n$ samples, $\hat{z} = (\hat{z}_1, \ldots, \hat{z}_n)$ is the estimated assignment, and $S_K$ is the set of all permutations of $\{1, \ldots, K\}$.

In the single-view case, we derived a lower bound on the misclustering error by introducing the advantage parameter for each cluster $a$ as $\delta_a = r_a - \max_{b \neq a} p_{ab}$ and the minimum advantage parameter as $\delta = \min_a \delta_a$, where $r_a$ is the probability that a point from cluster $a$ is correctly assigned and $p_{ab}$ is the probability that a point from cluster a is assigned to cluster b where b $\neq$ a.

This leads to the bound:

$$\mathbb{E}[h(\hat{z}, z^*)]_{\text{original view}} \geq \frac{1-\delta}{2},$$

which is valid beyond Gaussian mixtures.

We then analyzed multi-view consensus clustering, where $m$ independent transformed views are clustered separately and the final label is determined via majority vote. By applying Hoeffding's

inequality to the sum of correct votes across views, we derive the following upper bound as function of $\delta$ and m:

$$\mathbb{E}[h(\hat{z}, z^*)]_{\text{consensus}} \leq (K-1)\exp\left(-\frac{m\delta^2}{2}\right),$$

The upper bound shows exponential decay in $m$ whenever the per-view advantage $\delta$ is positive.

The main conclusion is that, while the single-view misclustering rate is bounded below by a positive constant, the multi-view consensus error decreases exponentially with the number of views $m$ under mild conditions on view diversity and informativeness. Therefore, there exists a finite $m_0$ such that for all $m > m_0$,

$$\mathbb{E}[h(\hat{z}, z^*)]_{\text{consensus}} < \mathbb{E}[h(\hat{z}, z^*)]_{\text{original view}}.$$

This establishes the theorem that consensus clustering with sufficiently many independent informative views strictly improves upon single-view clustering in expectation.

For a multivariate Gaussian, the covariance matrix fully determines the joint distribution. Thus, zero covariance implies that the joint density factorizes, and the variables are independent.

Moreover, one can argue that weakly uncorrelated views contribute proportionally to their degree of uncorrelation: a lower correlation implies that each view is more effective, and fewer views are required to achieve the same performance. Consequently, for weakly uncorrelated views, the total number of views $m$ is greater than or equal to the number of *effective* views.

---

**Theorem for Consensus Clustering**
**Condition 1 (View Diversity):** The collection $\{X_v\}_{v=1}^m$ is mutually independent.
**Condition 2 (View Informativeness):**

$$r_a^{(v)} > \max_{b \neq a} p_{ab}^{(v)} \quad \text{for all } a \in \{1, \ldots, K\}, v \in \{1, \ldots, m\}$$

which implies $\delta = \min_{v,a} \delta_a^{(v)} > 0$ where $\delta_a^{(v)} = r_a^{(v)} - \max_{b \neq a} p_{ab}^{(v)}$
**Result:**

$$\mathbb{E}[h(\hat{z}, z^*)]_{\text{consensus}} \leq (K-1)\exp\left(-\frac{m\delta^2}{2}\right) \xrightarrow{m \to \infty} 0$$

$$\exists m_0 \text{ such that } \forall m > m_0, \quad \mathbb{E}[h(\hat{z}, z^*)]_{\text{consensus}} < \mathbb{E}[h(\hat{z}, z^*)]_{\text{original view}}$$

---

### 2.3 LATENT SPACE REPRESENTATION LEARNING

After obtaining the refined cluster assignment, we train the MLP $q_\phi$ to produce cluster shaped latent representations. Our training objective is to maximize the mutual information between embeddings $\boldsymbol{h}_i \in \mathbb{R}^d$ and their assigned cluster centroids $\boldsymbol{c}_i \in \mathbb{R}^d$, to ensure that the learned representations reflect cluster-level semantics. Let $H$ and $C$ be the random variables corresponding to the embeddings and cluster centroids, respectively. The mutual information is defined as:

$$I(H;C) = H(C) - H(C \mid H) = -\sum_{k=1}^{K} \pi_k \log \pi_k + \frac{1}{N}\sum_{i=1}^{N}\sum_{k=1}^{K} \gamma_{ik} \log \gamma_{ik}. \tag{1}$$

Mutual information is maximized by (i) encouraging balanced cluster weights $\pi_k = 1/K$ to increase $H(C)$, and (ii) promoting confident assignments $\gamma_{ik} \to 1$ for the true cluster to reduce $H(C \mid H)$. Since changing $H(C)$ alters the cluster distribution, our practical optimization focuses on minimizing $H(C \mid H)$ by sharpening the assignment probabilities $\gamma_{ik}$ (see Appendix A for details).

The assignment probability is given by:

$$\gamma_{ik} = \frac{\pi_k \cdot \mathcal{N}(\boldsymbol{h}_i \mid \boldsymbol{\mu}_k, \sigma_k)}{\sum_{j=1}^{K} \pi_j \cdot \mathcal{N}(\boldsymbol{h}_i \mid \boldsymbol{\mu}_j, \sigma_j)}. \tag{2}$$

For L2-normalized embeddings, this reduces to a softmax over cosine similarities:

$$\mathcal{N}(\mathbf{h}_i \mid \boldsymbol{\mu}, \sigma^2) \propto \exp\left(\frac{\text{sim}(\mathbf{h}_i, \boldsymbol{\mu})}{\sigma^2}\right). \tag{3}$$

This directly connects to the InfoNCE loss, which for a single sample $\mathbf{h}_i$ is defined as:

$$\mathcal{L}_{\mathrm{N}}^{(i)} = -\log \frac{\exp\left(\frac{\mathrm{sim}(\boldsymbol{h}_i, \boldsymbol{\mu}_k)}{\tau}\right)}{\sum_{j=1}^{K} \exp\left(\frac{\mathrm{sim}(\boldsymbol{h}_i, \boldsymbol{\mu}_j)}{\tau}\right)}, \tag{4}$$

with the total loss averaged over all samples:

$$\mathcal{L}_{\mathrm{N}} = \frac{1}{N} \sum_{i=1}^{N} \mathcal{L}_{\mathrm{N}}^{(i)}. \tag{5}$$

Minimizing $\mathcal{L}_{\mathrm{N}}$ increases similarity between embeddings and their assigned centroids while decreasing similarity to others, thereby sharpening $\gamma_{ik}$, reducing $H(C \mid H)$, and consequently maximizing $I(H; C)$ (Appendix A).

On the other hand, our prior assumption is that embeddings are generated from a Gaussian Mixture Models. We enforce the Gaussian mixture prior through a negative log-likelihood loss:

$$\mathcal{L}_{\mathrm{GMM}} = -\sum_{i=1}^{n} \log \left(\sum_{k=1}^{K} \pi_k \mathcal{N}(\boldsymbol{h}_i | \boldsymbol{\mu}_k, \boldsymbol{\Sigma}_k)\right),$$

The combined optimization objective integrates both losses: $\mathcal{L} = \alpha \mathcal{L}_{\mathrm{InfoNCE}} + \beta \mathcal{L}_{\mathrm{GMM}}$,

Backpropagation through $\mathcal{L}$ simultaneously pulls embeddings toward assigned centroids while repelling others via $\mathcal{L}_{\mathrm{InfoNCE}}$ and constrains embeddings to lie on the manifold defined by the Gaussian mixture via $\mathcal{L}_{\mathrm{GMM}}$. Algorithm 2 describes the iterative algorithm that iterate between applying algorithm 1 for consensus clustering and training on joint contrastive and GMM negative log-likelihood losses.

---

**Algorithm 1** Consensus Multi-view Text Clustering

---

**Require:** Embeddings $H = \{\boldsymbol{h}_i\}_{i=1}^{n}$, number of views $m$, clusters $K$
**Ensure:** Consensus labels $\hat{y} \in \{1, \ldots, K\}^n$
1  Initialize transforms $\{T^{(v)}\}_{v=1}^{m}$ (e.g., PCA, encoders, perturbations)
2  **for** $v \leftarrow 1$ **to** $m$ **do**
3   $\quad H^{(v)} \leftarrow T^{(v)}(h),\ R^{(v)} \leftarrow \mathrm{GMM}(H^{(v)}, K)$          // soft $n \times K$ assignments
   $\quad c^{(v)} \leftarrow \arg\max R^{(v)}$                              // hard labels
4  $W_{ij} \leftarrow \frac{1}{m} \sum_v \mathbf{1}\{c_i^{(v)} = c_j^{(v)}\},\ L \leftarrow I - D^{-1/2} W D^{-1/2}$ with $D = \mathrm{diag}(W\mathbf{1}),\ U \leftarrow K$ smallest eigenvectors of $L$ row-normalized, $\hat{y} \leftarrow \mathrm{KMeans}(U, K)$
5  **return** $\hat{y}$

---

**Algorithm 2** Iterative Latent Space Learning with Consensus

---

**Require:** Embeddings $\{\boldsymbol{h}_i\}$, clusters $K$, encoder $\phi_\theta$, parameters $\{\pi_k, \boldsymbol{\mu}_k, \boldsymbol{\Sigma}_k\}$, weights $\alpha, \beta$, temperature $\tau$, epochs $E$, Number of epochs as Clustering Interval $e$
**Ensure:** Trained $\phi_\theta$, consensus assignments $\hat{y}$
6  **for** $epoch \leftarrow 1$ **to** $E$ **do**
7   $\quad$ **if** $epoch\%e = 0$ **then**
8    $\quad\quad$ Run Alg. 1, update $\hat{y}, \{\boldsymbol{\mu}_k\}$

9   $\quad$ **InfoNCE loss:** $\mathcal{L}_{\mathrm{InfoNCE}} = \frac{1}{N} \sum_i -\log \frac{\exp(\frac{\mathrm{sim}(\boldsymbol{h}_i, \boldsymbol{\mu}_{\hat{y}_i})}{\tau})}{\sum_j \exp(\frac{\mathrm{sim}(\boldsymbol{h}_i, \boldsymbol{\mu}_j)}{\tau})}$
10  $\quad$ **GMM loss:** $\mathcal{L}_{\mathrm{GMM}} = -\sum_i \log\left(\sum_k \pi_k \mathcal{N}(\boldsymbol{h}_i | \boldsymbol{\mu}_k, \boldsymbol{\Sigma}_k)\right)$
11  $\quad$ **Training:** $\mathcal{L} = \alpha \mathcal{L}_{\mathrm{InfoNCE}} + \beta \mathcal{L}_{\mathrm{GMM}} \Rightarrow$ Update encoder $\phi_\theta$ by backpropagation
12  **return** encoder $\phi_\theta$

---

## 3 EVALUATION

We conduct our experimentations using 2 datasets: DBPedia and Reuters R8 Datasets. We evaluate first the Consensus Clustering alone, then we evaluate the effectiveness of the iterative co-training by testing on unseen text data.

### 3.1 MULTI-VIEW CONSENSUS LEARNING EVALUATION

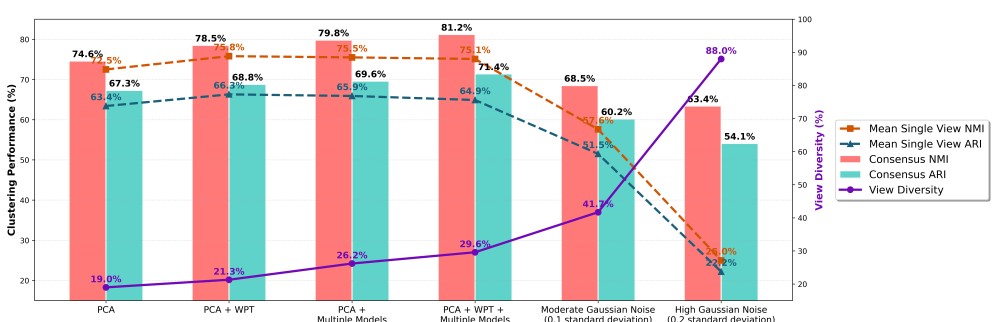

Figure 2: *Multi-view Clustering Performance against View Diversity*

Table 2: NMI and ARI values of several models under some transformations on the DBPedia Dataset with $k = 8$ clusters

| | Metric | Single View | | | | Multi-View Consensus Clustering | |
| | | KMeans | Gaussian Mixture Model | | Spectral Clustering | Gaussian Mixture Model | |
| | | | Heterogeneous | Homogeneous | | Heterogeneous | Homogeneous |
|---|---|---|---|---|---|---|---|
| **Original Embeddings** | NMI | 69.5 | 68.8 | 69.5 | 61.4 | – | – |
| | ARI | 60.6 | 60.3 | 60.7 | 52.5 | – | – |
| **WPT Transform** | NMI | – | $70.5 \pm 0.7$ | $73.6 \pm 2.3$ | – | 71.0 | **73.0** |
| | ARI | – | $60.7 \pm 1.4$ | $63.6 \pm 0.9$ | – | 61.7 | **63.2** |
| **PCA** | NMI | – | $70.1 \pm 0.5$ | $74.1 \pm 1.9$ | – | 70.6 | **74.6** |
| | ARI | – | $59.6 \pm 1.6$ | $65.5 \pm 1.8$ | – | 61.6 | **67.3** |
| **PCA + WPT Transform** | NMI | – | $65.0 \pm 6.3$ | $73.9 \pm 4.4$ | – | 71.4 | **78.8** |
| | ARI | – | $60.7 \pm 1.4$ | $63.6 \pm 0.9$ | – | 61.0 | **69.3** |
| **PCA + Gaussian Noise** | NMI | – | $68.3 \pm 1.7$ | $69.6 \pm 4.0$ | – | 70.8 | **76.0** |
| | ARI | – | $58.8 \pm 2.2$ | $61.0 \pm 4.1$ | – | 61.4 | **66.9** |
| **PCA + Multiple Models** | NMI | – | $71.0 \pm 2.2$ | $75.6 \pm 3.0$ | – | 72.6 | **78.9** |
| | ARI | – | $59.9 \pm 3.1$ | $66.2 \pm 3.6$ | – | 62.4 | **68.8** |
| **PCA + Gaussian Noise + Multiple Models** | NMI | – | $68.5 \pm 3.2$ | $70.8 \pm 4.8$ | – | 77.6 | **80.0** |
| | ARI | – | $57.8 \pm 3.8$ | $61.1 \pm 5.7$ | – | 66.7 | **70.0** |
| **PCA + WPT + Multiple Models** | NMI | – | $66.3 \pm 5.2$ | $73.4 \pm 4.8$ | – | 75.2 | **81.2** |
| | ARI | – | $53.5 \pm 6.6$ | $62.9 \pm 6.6$ | – | 65.0 | **71.4** |

Table 3: NMI and ARI values of several models under some transformations on the DBPedia Dataset with $k = 14$ clusters

| | Metric | Single View | | | | Multi-View Consensus Clustering | |
| | | KMeans | Gaussian Mixture Model | | Spectral Clustering | Gaussian Mixture Model | |
| | | | Heterogeneous | Homogeneous | | Heterogeneous | Homogeneous |
|---|---|---|---|---|---|---|---|
| **Original Embeddings** | NMI | 72.7 | 69.4 | 72.8 | 58.5 | – | – |
| | ARI | 61.2 | 57.5 | 61.2 | 46.2 | – | – |
| **WPT Transform** | NMI | – | $72.2 \pm 1.1$ | $76.7 \pm 1.2$ | – | 73.7 | **77.9** |
| | ARI | – | $59.5 \pm 1.7$ | $65.0 \pm 1.9$ | – | 62.0 | **67.5** |
| **Multiple Models** | NMI | – | $71.1 \pm 1.8$ | $75.1 \pm 2.2$ | – | 75.3 | **78.0** |
| | ARI | – | $58.8 \pm 2.3$ | $63.1 \pm 2.7$ | – | 63.1 | **65.9** |
| **PCA + Multiple Models** | NMI | – | $73.0 \pm 1.7$ | $75.9 \pm 2.1$ | – | 76.8 | **79.6** |
| | ARI | – | $58.9 \pm 2.4$ | $63.6 \pm 2.8$ | – | 63.4 | **67.7** |
| **PCA + Gaussian Noise + Multiple Models** | NMI | – | $70.8 \pm 2.5$ | $71.5 \pm 4.0$ | – | 77.2 | **79.4** |
| | ARI | – | $57.8 \pm 2.9$ | $60.1 \pm 4.0$ | – | 64.2 | **67.6** |
| **PCA + WPT + Multiple Models** | NMI | – | $66.4 \pm 6.7$ | $74.0 \pm 4.2$ | – | 76.8 | **80.8** |
| | ARI | – | $50.0 \pm 9.4$ | $59.6 \pm 7.6$ | – | 64.0 | **68.8** |

Table 4: NMI and ARI values of several models under some transformations on the Reuters R8 Dataset with k = 6 clusters

| | Metric | Single View | | | | Multi-View Consensus Clustering | |
|---|---|---|---|---|---|---|---|
| | | KMeans | Gaussian Mixture Model | | Spectral Clustering | Gaussian Mixture Model | |
| | | | Heterogeneous | Homogeneous | | Heterogeneous | Homogeneous |
| **Original Embeddings** | NMI | 72.8 | 73.9 | 65.0 | 68.6 | – | – |
| | ARI | 68.0 | 70.0 | 55.4 | 61.4 | – | – |
| **PCA + Gaussian Noise + Multiple Models** | NMI | – | 64.1 ±5.0 | 69.3 ±7.0 | – | 74.7 | **81.0** |
| | ARI | – | 62.6 ±5.4 | 63.1 ±8.8 | – | 71.9 | **75.0** |
| **PCA + WPT + Multiple Models** | NMI | – | 62.8 ±11.0 | 70.2 ±10.8 | – | 77.5 | **82.5** |
| | ARI | – | 57.7 ±13.7 | 64.1 ±14.7 | – | 73.7 | **81.1** |

We evaluate the Multi-view Consensus Clustering framework against standard baselines including K-Means, Gaussian Mixture Models (GMM), and Spectral Clustering. For both the GMM baseline and our consensus approach, we consider two covariance settings: isotropic heterogeneous (Case 1) and isotropic homogeneous (Case 2). Experimental results demonstrate that our consensus model achieves superior performance under homogeneous isotropic covariance. As shown in Tables 2, 3 and 4, our framework consistently outperforms baselines on both DBPedia and Reuters R8 datasets when combined with appropriate transformations. The method shows particular effectiveness with high perturbations (e.g., PCA + WPT + Multiple Models) that preserve semantic structure.

Figure 2 illustrates the relationship between the diversity of the generated views, the clustering performance of individual views, and the overall performance of the consensus method. Diversity is quantified as the mean ARI value across all pairwise combinations of the generated views. The results demonstrate that the effectiveness of consensus clustering is determined solely by two factors: the diversity among the generated views and the clustering performance of the Gaussian mixture model on these views. This empirical observation supports the theorem established earlier, which links the performance of consensus clustering to its ability to satisfy both the *diversity condition* and the *informativeness condition*. Specifically, the theoretical proof in the appendix showed that the consensus clustering error depends on the number of diverse views $m$ and on the advantage term $\delta$ of each view, the latter being directly related to the clustering quality of the corresponding transformed view. The plot confirms this by showing that the PCA + WPT + Multiple Models transformation achieves the best results, as it more closely satisfies the conditions of the theorem. These findings highlight the importance of transformation choice. Transformations that generate a greater degree of diversity (e.g., through stronger feature corruption or nonlinear perturbations) while still preserving the informativeness condition (Condition 2) are more likely to produce less correlated views. By effectively increasing the usable number of views $m$, such transformations enable the consensus method to achieve a lower error floor, thereby more accurately reflecting the theoretical guarantees of the proof.

## 3.2 TRAINING EVALUATION

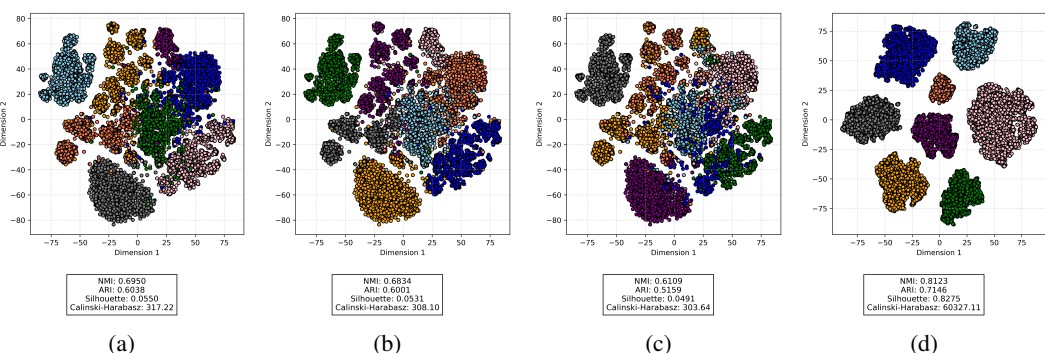

(a)  (b)  (c)  (d)

Figure 3: Comparative clustering results on the DBPedia dataset, for k = 8 clusters. a) KMeans Clustering. b) Gaussian Mixture Model Clustering. c) Spectral Clustering. d) Consensus Clustering - Contrastive Training at epoch 20.

We assess the effectiveness of the training procedure in reshaping the latent space such that it becomes more amenable to clustering. As illustrated in Figure 3, our method successfully brings embeddings closer to their assigned centroids, thereby validating its effectiveness. This observation is validated when looking at the silhouette score and the Calinski-Harabasz score. Both metrics indicate a more topologically clustered latent space, which can subsequently be more effectively clustered by a simple algorithm such as KMeans.

### 3.3 CLUSTERING ON UNSEEN TEST DATA

We investigate whether a clustering model trained on a subset of the data can be effectively applied to unseen samples, under the assumption that these samples are drawn from the same underlying distribution as the training data. The results, reported in Table 5 and Figure 4, indicate that clustering can be performed on a relatively small subset of the dataset, after which the trained model reshapes the latent space such that the resulting embeddings are more amenable to clustering using a simple KMeans or GMM algorithm.

Table 5: Clustering Performance under Different Settings, after 10 epochs

| Setting | NMI (Train) | ARI (Train) | NMI (Test) | ARI (Test) |
|---|---|---|---|---|
| 90% Training - 10% Testing | 80.4 | 71.1 | 79.6 | 70.3 |
| 50% Training - 50% Testing | 80.4 | 71.1 | 79.9 | 70.8 |
| 20% Training - 80% Testing | 80.1 | 68.7 | 78.7 | 67.6 |
| 10% Training - 90% Testing | 81.3 | 71.5 | 79.5 | 70.1 |

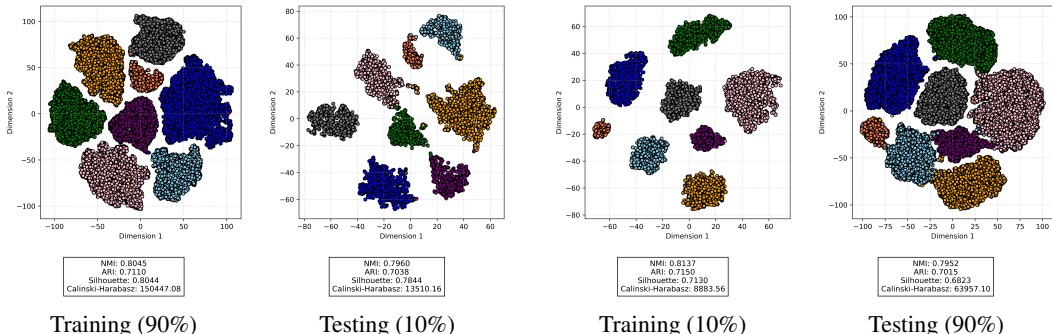

(a) Training on 90% and Testing on 10% of the text documents.

(b) Training on 10% and Testing on 90% of the text documents.

Figure 4: Clustering performance on unseen text documents from the DBPedia dataset for $k = 8$ clusters at epoch 10.

## 4 CONCLUSION

In this work, we present a consensus clustering method that generates multi-view transformations of the original embeddings to achieve a better clustering performance compared to single-view clustering. The clustering effectiveness increases with the degree of diversity, which is determined by the number of uncorrelated views generated by the transformation, and the clustering performance on each single view.

Then, we trained an MLP encoder to project the original high-dimensional latent space into a lower-dimensional representation, where applying KMeans clustering yields improved results. The combination of contrastive loss and Gaussian negative log-likelihood contributes to shaping a latent space that enhances clustering quality and maintains consistency with the Gaussian prior assumption.

Finally, we demonstrated that the proposed model generalizes effectively to unseen text documents, achieving robust clustering performance even when trained on a relatively small fraction of the available dataset.

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

## A  APPENDIX

### PROOF 1 - MINIMIZING INFONCE LOSS MAXIMIZES MUTUAL INFORMATION BETWEEN EMBEDDINGS AND THEIR ASSIGNED CLUSTER CENTROIDS

In this proof, we demonstrate that minimizing the InfoNCE loss maximizes the mutual information between embeddings and their assigned cluster centroids. The mutual information $I(\mathbf{h}; \mathbf{c})$ between node embeddings $\mathbf{h}$ and cluster assignments $\mathbf{c}$ is defined as:

$$I(\mathbf{h}; \mathbf{c}) = H(\mathbf{c}) - H(\mathbf{c} \mid \mathbf{h}) \tag{6}$$

where $H(\mathbf{c})$ is the entropy of cluster assignments:

$$H(\mathbf{c}) = -\sum_{k=1}^{K} p(c_k) \log p(c_k) = -\sum_{k=1}^{K} \pi_k \log \pi_k \tag{7}$$

and $H(\mathbf{c} \mid \mathbf{h})$ is the conditional entropy:

$$H(\mathbf{c} \mid \mathbf{h}) = -\sum_{i=1}^{N} \sum_{k=1}^{K} p(\mathbf{h}_i) p(c_k \mid \mathbf{h}_i) \log p(c_k \mid \mathbf{h}_i) = -\frac{1}{N} \sum_{i=1}^{N} \sum_{k=1}^{K} \gamma_{ik} \log \gamma_{ik}. \tag{8}$$

The mutual information is therefore expressed as:

$$I(\mathbf{h};\mathbf{c}) = -\sum_{k=1}^{K} \pi_k \log \pi_k + \frac{1}{N} \sum_{i=1}^{N} \sum_{k=1}^{K} \gamma_{ik} \log \gamma_{ik} \quad (9)$$

Mutual information maximization is achieved through two mechanisms:

1. Maximizing $H(\mathbf{c})$ by encouraging uniform cluster weights $\pi_k = \frac{1}{K}$;

2. Minimizing $H(\mathbf{c} \mid \mathbf{h})$ by promoting confident assignments where $\gamma_{ik} \to 1$ for the true cluster and $\gamma_{ik} \to 0$ for others.

We initialize cluster weights with a uniform distribution $\frac{1}{K}$ but training the optimal clustering might diverge the weights from being uniform, which is totally acceptable since our main objective is to yield the best possible clustering which is sometimes reached with non-uniform weight distribution. But since we also want to maximize $I(\mathbf{h}, \mathbf{c})$ and since $H(\mathbf{c})$ is not to be changed because it will directly affect the clustering, the objective of maximizing mutual information while preserving the optimal clustering would be to minimize $H(\mathbf{c}|\mathbf{h})$ by promoting confident assignments.

The soft assignment probability $\gamma_{ik}$ is computed as:

$$\gamma_{ik} = \frac{\pi_k \cdot \mathcal{N}(\boldsymbol{h}_i \mid \boldsymbol{\mu}_k, \sigma_k)}{\sum_{j=1}^{K} \pi_j \cdot \mathcal{N}(\boldsymbol{h}_i \mid \boldsymbol{\mu}_j, \sigma_j)} \quad (10)$$

which is maximized when $\mathcal{N}(\boldsymbol{h}_i \mid \boldsymbol{\mu}_k, \sigma_k)$ is maximized and $\mathcal{N}(\boldsymbol{h}_i \mid \boldsymbol{\mu}_j, \sigma_j)$ is minimized for all $j \neq k$.

For L2-normalized embeddings where $\|\mathbf{h}_i\| = \|\boldsymbol{\mu}\| = 1$:

$$\text{sim}(\mathbf{h}_i, \boldsymbol{\mu}) = \frac{\mathbf{h}_i^\top \boldsymbol{\mu}}{\|\mathbf{h}_i\| \, \|\boldsymbol{\mu}\|} = \mathbf{h}_i^\top \boldsymbol{\mu}$$

and:

$$\|\mathbf{h}_i - \boldsymbol{\mu}\|^2 = 2 - 2\mathbf{h}_i^\top \boldsymbol{\mu} = 2(1 - \text{sim}(\mathbf{h}_i, \boldsymbol{\mu})).$$

The Gaussian density simplifies to:

$$\mathcal{N}(\mathbf{h}_i \mid \boldsymbol{\mu}, \sigma^2) = \frac{1}{\sqrt{2\pi\sigma^2}} \exp\left(-\frac{\|\mathbf{h}_i - \boldsymbol{\mu}\|^2}{2\sigma^2}\right) = \frac{1}{\sqrt{2\pi\sigma^2}} \exp\left(\frac{\text{sim}(\mathbf{h}_i, \boldsymbol{\mu})}{\sigma^2}\right) \quad (11)$$

$$\mathcal{N}(\mathbf{h}_i \mid \boldsymbol{\mu}, \sigma^2) \propto \exp\left(\frac{\text{sim}(\mathbf{h}_i, \boldsymbol{\mu})}{\sigma^2}\right)$$

The InfoNCE loss for a single sample $\mathbf{h}_i$ is defined as:

$$\mathcal{L}_{\text{N}}^{(i)} = -\log \frac{\exp\left(\frac{\text{sim}(\boldsymbol{h}_i, \boldsymbol{\mu}_k)}{\tau}\right)}{\sum_{j=1}^{K} \exp\left(\frac{\text{sim}(\boldsymbol{h}_i, \boldsymbol{\mu}_j)}{\tau}\right)} \quad (12)$$

with the total loss averaged over all samples:

$$\mathcal{L}_{\text{N}} = \frac{1}{N} \sum_{i=1}^{N} \mathcal{L}_{\text{N}}^{(i)} \quad (13)$$

Minimizing $\mathcal{L}_{\text{N}}$ increases the similarity between $\mathbf{h}_i$ and its assigned centroid $\boldsymbol{\mu}_k$ while decreasing similarity to other centroids. This sharpens the posterior distribution $\gamma_{ik}$, reducing $H(\mathbf{c} \mid \mathbf{h})$ which maximizes the mutual information $I(\mathbf{h}; \mathbf{c})$ constrained on obtaining the optimal clustering.

PROOF 2 - CONSENSUS CLUSTERING ACHIEVES A LOWER EXPECTED MISCLUSTERING RATE THAN SINGLE VIEW CLUSTERING

In this proof, we demonstrate that running any clustering model on multiple transformed views of the data, followed by a spectral consensus step on a co-occurrence matrix, yields a strictly smaller expected misclustering rate than applying the same clustering procedure on a single view, under some conditions.

For any clustering algorithm, define:

- $r_a$ = probability that a point from cluster $a$ is correctly assigned
- $p_{ab}$ = probability that a point from cluster $a$ is assigned to cluster $b$ ($b \neq a$)

Define the advantage for cluster $a$ as:

$$\delta_a = r_a - \max_{b \neq a} p_{ab}.$$

This measures how much better the algorithm is at correct assignment vs. its highest misassignment.

Let $\delta = \min_a \delta_a$ be the minimum advantage across all clusters.

We define the misclustering fraction for an estimator $\hat{z}$ is:

$$h(\hat{z}, z^*) = \min_{\pi \in S_K} \frac{1}{n} \sum_{j=1}^{n} \mathbf{1}\{\pi(\hat{z}_j) \neq z_j^*\}.$$

where $z^* = (z_1^*, \ldots, z_n^*)$ denotes the true cluster assignment of all $n$ samples, $\hat{z} = (\hat{z}_1, \ldots, \hat{z}_n)$ is the estimated assignment, and $S_K$ is the set of all permutations of $\{1, \ldots, K\}$.

A - LOWER BOUND ON THE EXPECTED MISCLUSTERING RATE OF THE GAUSSIAN MIXTURE MODEL

Our goal is to derive a lower bound on the minimax risk:

$$\inf_{\hat{z}} \sup_{z^*} \mathbb{E}[h(\hat{z}, z^*)].$$

For each cluster $a$, we have:

$$1 - r_a = \sum_{b \neq a} p_{ab} \geq \max_{b \neq a} p_{ab},$$

and thus:

$$1 - \delta_a = 1 - r_a + \max_{b \neq a} p_{ab} \leq 2(1 - r_a),$$

which implies:

$$1 - r_a \geq \frac{1 - \delta_a}{2} \geq \frac{1 - \delta}{2}.$$

Now consider the worst-case scenario where only two clusters have the minimum advantage $\delta$ and the remaining clusters are perfectly separated ($\delta_a = 1$ for other clusters).

The error is lower bounded by the error on the two closest clusters:

$$\boxed{\mathbb{E}[h(\hat{z}, z^*)]_{\text{original view}} \geq \frac{1 - \delta}{2}}$$

The key insight is that there is a fundamental limit to how well any clustering algorithm can perform, determined by the advantage parameter.

## B. MULTI-VIEW CONSENSUS CLUSTERING IMPROVES UPON SINGLE-VIEW CLUSTERING

Suppose we have $m$ independent views of the data, each obtained by applying a transformation.

Assume that for each view $v$, the clustering algorithm produces labels such that for any point from cluster $a$:

$$r_a^{(v)} > \max_{b \neq a} p_{ab}^{(v)}.$$

Define the advantage for cluster $a$ in view $v$ as:

$$\delta_a^{(v)} = r_a^{(v)} - \max_{b \neq a} p_{ab}^{(v)}.$$

Let

$$\delta^{(v)} = \min_a \delta_a^{(v)}, \quad \text{and} \quad \delta = \min_v \delta^{(v)}.$$

We assume $\delta > 0$.

The views are independent. For multi-view consensus clustering, we consider majority voting: each point is assigned to the cluster that wins the majority of votes across views.

For a fixed point from cluster $a$, the probability that it is misclassified to cluster $b$ is the probability that the number of views assigning it to $b$ is at least the number assigning it to $a$. Let $X_v$ be the indicator that view $v$ assigns the point to $a$, and $Y_v$ be the indicator that view $v$ assigns it to $b$. Note that for each view, $X_v$ and $Y_v$ are not independent, but across views they are independent. Consider the difference $Z_v = X_v - Y_v$. Then the event that the point is assigned to $b$ rather than $a$ requires that the sum of $Z_v$ over $v$ is $\leq 0$. The expected value of $Z_v$ is $r_a^{(v)} - p_{ab}^{(v)} \geq \delta_a^{(v)} \geq \delta$. Applying Hoeffding's inequality to the sum of $Z_v$ (note that $Z_v$ takes values in $[-1, 1]$ with range 2), we get:

$$\Pr\left(\sum_{v=1}^{m} Z_v \leq 0\right) \leq \exp\left(-\frac{2(m\delta)^2}{m \cdot (2)^2}\right) = \exp\left(-\frac{m\delta^2}{2}\right).$$

By union bound over all $b \neq a$, the probability that a specific point from cluster $a$ is misclassified is at most:

$$\Pr(\text{error for point from } a) \leq (K - 1) \exp\left(-\frac{m\delta^2}{2}\right).$$

Since this bound holds for every point regardless of its cluster membership, the expected misclustering fraction is also bounded by:

$$\boxed{\mathbb{E}[h(\hat{z}, z^*)]_{\text{consensus}} \leq (K - 1) \exp\left(-\frac{m\delta^2}{2}\right).}$$

**Superiority of Multi-View Consensus**

From Part A, the expected misclustering error of a single view is bounded below by:

$$\mathbb{E}[h(\hat{z}, z^*)]_{\text{original view}} \geq \frac{1 - \delta}{2}$$

The Multi-view bound decays exponentially with $m$, while the single-view error is constant in $m$. Therefore, for sufficiently large $m$:

$$\boxed{\mathbb{E}[h(\hat{z}, z^*)]_{\text{consensus}} < \mathbb{E}[h(\hat{z}, z^*)]_{\text{original view}}}$$

**Conditions:**

**Condition 1 (View Diversity):** For any additional view $m$, the view must be independent of all previous views. Formally, the collection of indicator random variables $\{X_v\}_{v=1}^{m}$ is mutually independent, This ensures the new view provides non-redundant information essential for Hoeffding's inequality.

**Condition 2 (View Informativeness):** Each view must provide meaningful clustering information. Formally, for each true cluster $a$ and view $v$, the probability of correct assignment must exceed the maximum probability of incorrect assignment:

$$r_a^{(v)} > \max_{b \neq a} p_{ab}^{(v)} \quad \text{for all } a \in \{1, \ldots, K\}, v \in \{1, \ldots, m\}.$$

This implies the advantage parameter $\delta = \min_{v,a} \delta_a^{(v)} > 0$, where $\delta_a^{(v)} = r_a^{(v)} - \max_{b \neq a} p_{ab}^{(v)}$.

Under these conditions, the consensus misclustering error decays exponentially with the number of views $m$:

$$\mathbb{E}[h(\hat{z}, z^*)]_{\text{consensus}} \leq (K - 1) \exp\left(-\frac{m\delta^2}{2}\right).$$

Consequently:

$$\lim_{m \to \infty} \mathbb{E}[h(\hat{z}, z^*)]_{\text{consensus}} = 0.$$

Since the single-view misclustering error is bounded below by a positive constant $\frac{1-\delta}{2}$ (from Part A), there exists a finite $m_0$ such that for all $m > m_0$:

$$\mathbb{E}[h(\hat{z}, z^*)]_{\text{consensus}} < \mathbb{E}[h(\hat{z}, z^*)]_{\text{original view}}$$

That is, consensus clustering with sufficiently many views outperforms single-view clustering.

---

**Theorem for Consensus Clustering**
**Condition 1 (View Diversity):** The collection $\{X_v\}_{v=1}^m$ is mutually independent.
**Condition 2 (View Informativeness):**

$$r_a^{(v)} > \max_{b \neq a} p_{ab}^{(v)} \quad \text{for all } a \in \{1, \ldots, K\}, v \in \{1, \ldots, m\}$$

which implies $\delta = \min_{v,a} \delta_a^{(v)} > 0$ where $\delta_a^{(v)} = r_a^{(v)} - \max_{b \neq a} p_{ab}^{(v)}$
**Result:**

$$\mathbb{E}[h(\hat{z}, z^*)]_{\text{consensus}} \leq (K - 1) \exp\left(-\frac{m\delta^2}{2}\right) \xrightarrow{m \to \infty} 0$$

$$\exists m_0 \text{ such that } \forall m > m_0, \quad \mathbb{E}[h(\hat{z}, z^*)]_{\text{consensus}} < \mathbb{E}[h(\hat{z}, z^*)]_{\text{original view}}$$

---

