# OpenReview forum: "Beyond Single Views: Achieving Significant Gains in Text Clustering via Informative Diversification"
_ICLR.cc/2026/Conference — ICLR 2026 Conference Withdrawn Submission_

### Official Review · Reviewer_Az8T · 2025-10-21

**Soundness:** 2
**Presentation:** 2
**Contribution:** 2
**Rating:** 2
**Confidence:** 4

**Summary:**

This paper introduces a new multi-view consensus clustering framework for text data, designed to overcome the limitations of single-view clustering methods. It leverages informative diversification—creating multiple, semantically varied versions of embeddings—to achieve lower misclustering error and higher robustness.

The paper theoretically proves that, in multi-view consensus clustering, the expected misclustering error decays exponentially with the number of views (m), under diversity and informativeness conditions, whereas single-view clustering retains a positive lower bound on the misclustering rate.

The proposed methodology is three-fold: 1. Multi-view generation;
2.Consensus clustering, and 3. Latent space learning.  Experimentally,
the proposed consensus clustering consistently outperforms baseline
methods such as K-Means, single-view GMM, and Spectral Clustering.

**Strengths:**

1. The paper introduces a novel approach that integrates multiple
semantically diverse embeddings (“views”) into a spectral consensus
clustering framework. This idea—aggregating information from
different embedding transformations—represents a creative extension
of ensemble learning to modern text clustering, improving stability
and robustness over single-view methods.


2. It provides formal proofs showing that the expected misclustering
error decreases exponentially with the number of independent,
informative views.  The derivations connect clustering performance
with statistical guarantees, offering a clear theoretical
justification for why and when multi-view consensus is superior.


3. The method elegantly combines contrastive learning (InfoNCE loss) with
Gaussian Mixture Modeling (GMM) in a joint optimization loop.  This
hybrid objective balances representation learning and cluster density
modeling, yielding embeddings that are semantically rich.


4. Experiments show consistent improvements over baseline methods
(K-Means, GMM, Spectral Clustering) across multiple configurations.
The model maintains robust clustering accuracy on unseen data,
demonstrating generalization beyond the training set—a key challenge
in unsupervised learning.


5. The algorithms (Algorithm 1 and 2) are clearly described,
step-by-step, with well-defined mathematical notation and transparent
design choices (e.g., transformation types, consensus computation).
The combination of deterministic and stochastic transformations (like
PCA, WPT, Gaussian noise) provides practical reproducibility for
future studies.

**Weaknesses:**

1. The evaluation is narrow, using only two clean English datasets
(DBPedia and Reuters R8). There’s no evidence of scalability to
large, noisy, or multilingual corpora, nor any analysis of
computational cost.



2. The proof of exponential error reduction assumes that the multiple
views are independent and informative. In reality, the generated views
(e.g., PCA or similar BERT models) are highly correlated, so these
conditions are unlikely to hold.


3. The method is only compared against basic clustering algorithms
(K-Means, GMM, Spectral Clustering), omitting modern deep or
contrastive clustering baselines. There is also no ablation or
sensitivity analysis to show which components truly drive the
improvement. This limitation appears in the short length of the reference list.
Clustering is one of the most extensively studied areas in machine
learning, and any new significant proposal should relate to much more
recent related work—most of which this paper ignores.




4. "Informative diversification” is not formally defined or adaptively
measured; the approach relies heavily on pretrained embeddings without
clarifying how much the gains come from the transformations versus the
base models. This weakens interpretability and reproducibility.

**Questions:**

1. The paper only benchmarks against K-Means, GMM, and Spectral
Clustering. How would it perform against modern methods like DEC,
IDEC, SCAN, or graph-based and contrastive clustering models that
already integrate multiple representations?


2. The framework requires multiple clustering runs and spectral
decomposition steps. How well does it scale with large corpora or
high-dimensional embeddings, especially when the number of views (m)
increases?


3. Since the model relies heavily on high-quality sentence embeddings
(e.g., from BERT), are the observed gains primarily due to the
diversification strategy, or simply from strong pretrained
representations?



4. The theoretical guarantees rely on mutually independent and
informative views. However, deterministic transformations (e.g., PCA
or similar BERT encoders) are highly correlated. Can the claimed
exponential error decay still hold when view independence is violated?

---

> ### Author Response · Authors · 2025-11-26
> **We thank the reviewer for their thoughtful and detailed comments. Below we address the main concerns.**
>
> 1. **Benchmarking and comparison with recent methods.**
>    We agree that the original version (two clean datasets plus basic baselines) was too narrow. As described in the **Global Response**, we now evaluate on five standard benchmarks (Reuters, DBpedia, AG News, Yahoo Answers, SearchSnippets) and compare against 12 recent deep text clustering methods (2020–2025), using the authors’ reported scores where available. The Related Work section has been updated accordingly. Our consensus+latent-space pipeline remains competitive with these recent methods while keeping the overall architecture simple and easy to deploy.
>
> 2. **Scalability and computational cost.**
>    The main expensive component is the spectral consensus on the $N \times N$ co-occurrence matrix, which we perform only once at *training* time to align cluster labels and learn a reshaped latent space. At inference time, we neither regenerate all $m$ views nor re-run spectral clustering: we cluster in the learned latent space with a single GMM, at cost comparable to a standard single-view pipeline. A more detailed complexity discussion is given in the **Global Response**.
>
> 3. **Role of informativeness vs. diversification.**
>    Using a strong pre-trained encoder together with a GMM helps to satisfy the informativeness condition: each view, on average, prefers the true class over any rival. If this condition were violated, consensus learning would not perform well. However, the improvements we report arise from the combination of both informativeness and diversification (between views), as formalised in our theoretical analysis.
>
> 4. **Independence, correlated views, and exponential error decay.**
>    We fully agree that deterministic transformations (PCA, WPT, related BERT encoders) produce correlated views, so the independent-view assumptions of the original Proof 2-B do not accurately describe our experimental setting. In the revised version (see **Global Response**), we explicitly treat Proof 2-B as an ideal benchmark and introduce a new Proof 2-C that allows arbitrary dependence and defines an effective number of views $m_{\mathrm{eff}}$ (non-degenerate views with positive average conditional margin). We obtain the bound
>    $E[h(\hat{z}, z^*)]{\text{consensus}} \le (K-1)\exp(- m_{\mathrm{eff}} \bar{\delta}^{2} / 8)$,
>    so exponential decay is in terms of $m_{\mathrm{eff}}$, not the raw $m$. When views become highly correlated, $m_{\mathrm{eff}}$ saturates and the bound no longer forces the error to vanish, which matches the empirical plateau we observe as more views are added.

---

### Official Review · Reviewer_CU4P · 2025-10-21

**Soundness:** 2
**Presentation:** 3
**Contribution:** 1
**Rating:** 2
**Confidence:** 4

**Summary:**

The paper proposes a new approach to do unsupervised-text-clustering. Particularly, they introduce the usage of multi-views combined with Gaussian Mixture Model to aggregate the views information.

**Strengths:**

The paper has clear writing as well as the reasoning process. I also find the claims made in the paper reasonable. While I have not rigorously validated the theoretical claims of the paper (in the appendix), I think they make sense intuitively on a high-level, thus, I believe they are correct.

**Weaknesses:**

First, I find the benchmark is lacking. The paper only evaluates their methods on 2 datasets. Furthermore, as mentioned in the related works, there are many other methods for text clustering and they only compare the proposed method to K-Mean, GMM, and Spectral clustering. Thus, I find the current benchmarking is not satisfiable for ICLR.

Second, while I believe the theoretical claims are correct, I question the assumptions that the authors use to make it works. For instance, the "mutual independent" (line 235) condition is too strong as I do not believe it ever holds in practice for a reasonable number of views m > 2. From my understanding, as all the views are generated on one entity/data point, it is much more likely for the views to be strongly dependent. On the other hand, I find the second condition "informative" is stated in a very unintuitive manner (overly formulated to fit the theoretical claim?). I also think it is difficult to validate how the assumption can hold in practice. Thus, I do not think it makes sense to say they are "mild conditions" as stated at line 224.

**Questions:**

See weakness

---

> ### Author Response · Authors · 2025-11-26
> **We thank the reviewer for their thoughtful and detailed comments. Below we address the main concerns.**
>
> 1. **Benchmarking and comparison with stronger methods.**
>    We fully agree that the original evaluation (two datasets, mainly traditional baselines) was not sufficient. In the revised version, we now evaluate on five standard benchmarks (Reuters, DBpedia, AG News, Yahoo Answers, SearchSnippets) and add comparisons with 12 recent deep text clustering methods (2020–2025), using the authors’ reported scores when available. These changes and the new comparison table are described in the **Global Response**.
>
> 2. **Independence, dependence, and effective number of views.**
>    We agree that strict mutual independence is unrealistic when all views are derived from the same texts. As clarified in the **Global Response**, we now explicitly treat the independent-view setting in Proof 2-B as an ideal benchmark regime, and introduce a new Proof 2-C that handles correlated views using a martingale argument and an effective number of views $m_{\mathrm{eff}}$.
> The resulting bound depends on $m_{\mathrm{eff}}$ rather than on the raw $m$, which explains the empirical plateau once additional views become highly correlated and effectively redundant.
>
> 3. **Informativeness condition (weak-learner assumption).**
>    The informativeness condition is now phrased more intuitively: each view, on average, prefers the true class over any rival by at least a small positive margin. Formally, for a point from class $a$ and view $v$ we require
>    $\mathbb{E}[Z_v \mid a] = r_a^{(v)} - \max_{b \neq a} p_{ab}^{(v)} \ge \delta > 0$,
>    so that each view behaves as a weak learner.
>
>    In the martingale-based analysis for correlated views, this becomes a condition on the *average* conditional margin of the effective views:
>    $\frac{1}{m_{\mathrm{eff}}}\sum_{v \in I_{\mathrm{eff}}} \mathbb{E}[Z_v \mid F_{v-1}, a] \ge \bar{\delta} > 0$,
>    where $I_{\mathrm{eff}}$ is the set of effective views and $m_{\mathrm{eff}} = |I_{\mathrm{eff}}|$ is its cardinality, and $F_{v-1}$ is the sigma-field generated by previous votes. We no longer describe these as “mild” conditions, but rather as a standard weak-learner assumption that can be checked empirically by verifying that individual views perform better than random.

---

### Official Review · Reviewer_MMDR · 2025-10-27

**Soundness:** 3
**Presentation:** 3
**Contribution:** 2
**Rating:** 4
**Confidence:** 4

**Summary:**

Briefly summarize the paper and its contributions. You can incorporate Markdown and Latex into your review. See https://openreview.net/faq.
The paper proposes a novel clustering scheme based on multi-view consensus clustering and dual-objective latent space optimization. Multi-view consensus is established by 1) first applying a view-specific randomly parameterized transforms, 2) conducting clustering for each view with Gaussian Mixture Model (GMM) clustering, and finally 3) merging the views via spectral clustering to yield the consensus cluster. Dual-objective latent space optimization refines the encoder latent space by minimizing both 1) an InfoNCE objective to sharpen the cluster assignment and 2) a regularization prior to map sample encodings onto the GMM manifold. The entire clustering process is iteratively conducted through alternation between the two processes. The authors further presents mathematical proof based on Hoeffding's inequality to demonstrate exponential error decrease induced by multi-view clustering.

**Strengths:**

In general, the presentation of the manuscript is good, and the proofs and methods are sound.

(Quality) The authors present theoretical backing to their scheme through discussions on the lowering of expected error caused by multi-view representation.

(Quality) Overall, the alternating optimization scheme proposed is interesting and methodologically sound.

(Quality) The authors demonstrate that their experiment outperforms single-view clustering.

(Clarity) The authors have presented the necessary equations and pseudocode to ensure a clear understanding for the audience.

**Weaknesses:**

In general, the work has issues with novelty and motivation, caused by the lack of a Related Work section and (more importantly) analysis of more recent literature (within the past 3 years).

(Quality) Experiments for clustering on unseen data are insufficient, as no single-view methods are presented for comparison.

(Originality) The work is somewhat limited in originality. Works on multi-view clustering with GMM [1] and InfoNCE [2] are already well-known. Thus, the proposed work incrementally builds upon existing work, with the main contribution being its application to LLM.

(Significance) The motivation of this work is detracted by the references selected. Aside from lacking a Related Works section, most of the references in the introduction are over 5 years old. An analysis of more recent multi-view clustering work (i.e. [3]) is necessary.

[1] Kumar, A., & Daumé, H. (2011). A co-training approach for multi-view spectral clustering. In Proceedings of the 28th international conference on machine learning (ICML-11) (pp. 393-400).

[2] Oord, A. V. D., Li, Y., & Vinyals, O. (2018). Representation learning with contrastive predictive coding. arXiv preprint arXiv:1807.03748.

[3] Pattnaik, A., George, C., Tripathi, R., Vutla, S., & Vepa, J. (2024, November). Improving hierarchical text clustering with llm-guided multi-view cluster representation. In Proceedings of the 2024 Conference on Empirical Methods in Natural Language Processing: Industry Track (pp. 719-727).

**Questions:**

1. Table 1 shows several sentence embedding models for generating multi-view representations. But their application is unclear. Does each model replace/merge its representation along with the Sentence-Bert output?

2. Given that multiple transformations are applied, would the cost (i.e. runtime, compute, memory) also increase significantly?

3. How many views m are used for the different multi-view schemes?

**Details Of Ethics Concerns:**

None.

---

> ### Author Response · Authors · 2025-11-26
> **We thank the reviewer for their thoughtful and detailed comments. Below we address the main concerns.**
>
> 1. **Related work and positioning.**
>    In the revised manuscript, we add a more recent Related Work section covering several deep and multi-view text clustering methods (2020–2025). As discussed in the **Global Response**, our goal is not to introduce yet another large deep architecture, but to study—both theoretically and empirically—the effect of diversification through consensus clustering. The new comparison table shows that our simple pipeline is competitive with recent deep clustering methods.
>
> 2. **Clustering on unseen data.**
>    For clustering on unseen data, we only use a single-view GMM at inference, applied to the reshaped latent space. This latent space has been trained using consensus clustering and a joint InfoNCE + GMM objective, so a simple GMM with few EM iterations performs much better than GMM on the original embeddings. In other words, the extra training cost is compensated by very cheap and effective inference.
>
> 3. **Originality of the method.**
>    We do not claim novelty for InfoNCE, GMM, or multi-view clustering individually. The contribution lies in their combination into a simple workflow:
>    (i) start from strong SBERT embeddings;
>    (ii) generate semantic-preserving but diverse views;
>    (iii) aggregate them via spectral co-occurrence consensus; and
>    (iv) learn a cluster-friendly latent space with a joint InfoNCE + GMM log-likelihood loss.
>    We then show empirical gains over single-view baselines and provide theoretical guarantees for the consensus step (see **Global Response**).
>
> 4. **Use of models in Table 1.**
>    The sentence embedding models listed in Table 1 are used to generate several embeddings for the same text. For each view, we randomly choose one of these base embeddings and then apply additional transformations (PCA, WPT, Gaussian noise). We clarify this procedure in the revised text.
>
> 5. **Cost of multiple transformations and scalability.**
>    On the cost side, the dominant component is the spectral consensus step on the $N \times N$ co-occurrence matrix, which is performed only at *training* time. At *inference* time, we do not regenerate all views or run spectral clustering: we simply apply a single GMM on the learned latent representation, with cost comparable to standard single-view clustering. EM also converges in fewer iterations thanks to the InfoNCE + GMM training. The full complexity analysis is given in the **Global Response**.
>
> 6. **Hyperparameters.**
>    To improve transparency and reproducibility, we now include a table in the revised manuscript with all key hyperparameters used in our experiments (e.g., $m = 40$, $\alpha = 1$, $\beta = 0.001$, $\tau = 0.05$).

---

### Official Review · Reviewer_8qY8 · 2025-10-28

**Soundness:** 2
**Presentation:** 2
**Contribution:** 2
**Rating:** 2
**Confidence:** 3

**Summary:**

This paper proposes a new text clustering method based on multi-view clustering and information diversification. The authors introduce an iterative framework that alternately refines the clustering step and the text representation learning step by jointly minimizing a combined loss function consisting of a likelihood term and an InfoNCE term. Multiple feature extraction methods and models are employed to obtain diverse views of the texts, and a voting mechanism is used to measure how many views support that any two texts belong to the same cluster. A spectral clustering algorithm is then applied to this voting matrix (an NxN similarity matrix, where N is the number of texts) to produce the final clustering result. The authors also provide a theoretical analysis explaining why multi-view clustering outperforms single-view clustering, based on the minimax risk of the misclassification rate. Experimental results demonstrate that multi-view clustering with information diversification yields significant performance improvements over single-view embeddings.

**Strengths:**

1. The paper is well written and easy to follow.

2. The experimental results are positive and provide support for the authors’ claims.

3. The proposed algorithms are simple, reproducible, and conceptually sound.

**Weaknesses:**

1. **Lack of comparison with stronger baselines.** The proposed method is compared only against standard embeddings (e.g., SBERT) combined with conventional clustering algorithms such as KMeans and GMM, as well as its own variants. However, comparisons with existing multi-view clustering methods or ensemble-based approaches are missing.

2. **Theoretical contribution appears limited.** The authors show that multi-view consensus clustering achieves an arbitrarily low minimax risk as the number of views increases, whereas single-view clustering retains a constant lower bound. This result, however, is rather straightforward and well known in ensemble learning theory. Moreover, the assumption that the multiple views are sufficiently diversified to achieve a large effective number of independent views is quite strong, making the comparison with the single-view case somewhat unfair.

3. **Limited novelty.** The main contribution—combining multi-view clustering with contrastive learning for text clustering—appears incremental relative to prior work.

4. **Questionable scalability.** The proposed algorithm involves computing eigenvalues during the spectral clustering step on an N×N matrix, where N is the number of texts. While this matrix can be stored sparsely, the voting mechanism across multiple diversified views may considerably reduce its sparsity, making the spectral clustering step computationally expensive. More discussion on scalability is warranted.

**Questions:**

1. How should Figure 2 and Table 2 be interpreted? In Table 2, performance drops after adding Gaussian noise, whereas Figure 2 suggests the opposite trend. Please clarify this inconsistency.

2. In Table 2 (first row), what exactly are the “original embeddings”? Which SBERT model was used to produce them, or were they obtained by concatenating embeddings from multiple models?

3. Are there other possible ways to combine multiple views besides the proposed consensus approach (voting + spectral clustering)?

---

> ### Author Response · Authors · 2025-11-26
> **We thank the reviewer for their thoughtful and detailed comments. Below we address the main concerns.**
>
> 1. **Experimental scope and deep baselines.**
>    In the revised version, we extend the evaluation from two to five benchmarks (Reuters, DBpedia, AG News, Yahoo Answers, SearchSnippets) and compile NMI/ARI/ACC scores for 12 recent deep text clustering methods (2020–2025) alongside our consensus work. The table shows that our pipeline is competitive with more complex architectures, despite its simplicity (see **Global Response**).
>
> 2. **Theory and the role of dependence.**
>    Our goal is not to claim novelty for the InfoNCE–mutual-information link, but to specialise ensemble-style guarantees to deep unsupervised text clustering and to clarify the role of dependence. Proof 2-B is now explicitly presented as an ideal independent-view benchmark. The revised manuscript adds a martingale-based Proof 2-C that replaces the raw number of views $m$ by an effective number $m_{\mathrm{eff}}$ of informative, non-degenerate views, and shows: $\mathbb{E}[h(\hat{z},z^*)]{\mathrm{consensus}} \le (K-1)\exp\bigl(-m_{\mathrm{eff}}\bar{\delta}^{2}/8\bigr)$,
>
>    This makes precise how correlated views reduce the effective ensemble size and explains the empirical plateau as $m$ grows (see **Global Response** for the full derivation).
>
> 3. **Novelty and positioning.**
>    Our goal is not to introduce yet another heavy deep architecture, but to systematically study the advantage of multi-view consensus clustering over single-view clustering. Our framework combines a simple consensus mechanism with a cluster-friendly latent-space training stage. This combination is, to the best of our knowledge, novel, remains easy to implement, and generalises well to unseen text. It provides both theoretical and empirical evidence of the gains brought by consensus clustering, and an efficient way to avoid recomputing consensus at inference time by reshaping the latent space via contrastive and GMM log-likelihood objectives (see **Global Response**).
>
> 4. **Computational cost and scalability.**
>    Regarding the computational analysis, the dominant computational bottleneck in our framework is the spectral consensus step on the $N \times N$ co-occurrence matrix. This step is performed only once at *training* time to obtain the final consensus clustering and shape the latent space. At *inference* time, we neither regenerate all views nor rerun spectral clustering: we simply cluster in the learned latent space with a single GMM, at a cost comparable to a standard single-view pipeline (see **Global Response**).
>
> 5. **Interpretation of Figure 2 and Table 2.**
>    Table 2 shows that adding a *small* amount of Gaussian noise to the embeddings can slightly help the consensus objective, because it increases diversity between views without severely affecting the informativeness condition. In contrast, Figure 2 explores regimes where the injected Gaussian noise becomes large enough that the embeddings themselves turn uninformative: diversity is then very high, but the informativeness condition is strongly violated and performance drops. This contrast also explains why the combination PCA + WPT + several SBERT models achieves the best results: it offers a favourable trade-off between diversity and informativeness.
>
> 6. **“Original embeddings” in Table 2.**
>    To generate the “original embeddings” in Table 2, we use the first SBERT model listed in Table 1, namely `all-MiniLM-L6-v2`. No concatenation across multiple encoders is involved.
>
> 7. **Other ways to combine multiple views.**
>    Once the co-occurrence matrix is constructed, any graph-based clustering method could in principle be applied on top of it. Some prior work instead learns a joint feature representation from multi-view feature extractions and then applies a single-view clustering algorithm on that consensus representation. In our case, we deliberately keep the combination simple: we leverage pretrained SBERT encoders plus lightweight transformations (PCA, WPT, Gaussian noise) and focus on majority voting across views.

---

> > ### Comment · Reviewer_8qY8 · 2025-11-26
> > **Thank you for the clarification**
> >
> > Thank you for the detailed rebuttal and for addressing my earlier questions. I have carefully reviewed your responses as well as the replies to the other reviewers.
> >
> > While the additional information is appreciated, it reinforces my original concerns. In particular, the key issues related to the theoretical novelty and practical applicability remain insufficiently resolved. For this reason, I have decided to maintain my original rating.

---

### Official Review · Reviewer_bHgA · 2025-10-30

**Soundness:** 2
**Presentation:** 3
**Contribution:** 1
**Rating:** 2
**Confidence:** 4

**Summary:**

This paper introduces a text clustering framework based on informative diversification, supported by a theoretical analysis of its multi-view consensus mechanism. While the reported gains over baselines such as K-Means merit recognition, the work's overall impact is limited by the restrictive nature of its theoretical assumptions and a lack of comprehensive experimental validation, which collectively undermine the validity of its claims.

**Strengths:**

1. Technical Framework: The paper presents a text clustering framework that integrates multi-view consensus clustering with deep representation learning in an end-to-end manner. An iterative co-training strategy is used to jointly optimize view generation, consensus clustering, and representation learning.

2. Theoretical Contributions: Theoretical analysis is provided, including an exponential upper bound on the misclustering error in multi-view consensus. The authors also connect the minimization of InfoNCE loss to the maximization of mutual information, giving a theoretical basis for the objective.

3. Experimental Results: Experiments on DBPedia and Reuters R8 report improvements in NMI and ARI over several baselines. The model shows some generalization capability, and t-SNE visualizations suggest that the learned embeddings form relatively compact clusters.

**Weaknesses:**

1. Limited Theoretical Novelty and Strong Assumptions: Proof 1 restates the known connection between InfoNCE and mutual information, contributing minimal theoretical innovation. Proof 2 relies on the strong—and often impractical—assumption of view independence, yet fails to discuss its validity or consequences in real-world scenarios.

2. Narrow Experimental Scope:  Evaluation is limited to two clean datasets and traditional baselines, lacking tests under noisy conditions or comparisons with recent deep learning-based clustering methods.

3. Insufficient Computational Analysis: The paper does not address the computational cost of multi-view generation or scalability, leaving practical feasibility unclear for larger datasets.

4. Incomplete Ablation and Hyperparameter Analysis: Ablation studies only explore view combinations without justifying core design decisions (e.g., spectral clustering vs. majority voting). Key hyperparameters—such as the number of views, loss weights, and iteration counts—lack systematic analysis, hindering reproducibility and insight.

5. Lack of Experimental Rigor and Presentation Issues: Critical implementation details and hyperparameter settings are inadequately documented. Additionally, Figure 2 uses inconsistent plot types for the same metric, impairing readability, and references are not alphabetized, reflecting a lack of attention to presentation quality.

**Questions:**

1.	The independence assumption for multiple views in Proof 2 is critical yet often impractical. How would violations of this assumption (i.e., correlated views) affect your theoretical guarantees, and did you empirically measure the dependence between the generated views?
2.	The experimental comparisons are limited to traditional methods. To firmly establish the advancement of your work, could you include results comparing against recent deep learning-based clustering approaches?
3.	Could you provide an analysis of the computational cost (e.g., training time scaling with dataset size and number of views) and include in the appendix the detailed settings for key hyperparameters (e.g., α, β, τ) to ensure reproducibility?

---

> ### Author Response · Authors · 2025-11-26
> **We thank the reviewer for their thoughtful and detailed comments. Below we address the main concerns.**
>
> 1. **Theoretical proofs and role of dependence.**
>    We fully agree that the InfoNCE–mutual-information link used in Proof 1 is classical and not meant as a stand-alone contribution. Our main theoretical contribution is twofold:
>    (i) we specialise ensemble-style guarantees to deep *unsupervised text* clustering via the per-view quantities $r_a^{(v)}$ and $p_{ab}^{(v)}$; and
>    (ii) more importantly, we extend the analysis beyond the ideal independent-view regime.
>
>    In the revised version, this is shown in the martingale-based Proof 2-C and the use of an *effective* number of views $m_{\mathrm{eff}}$, yielding the bound: $\mathbb{E}[h(\hat{z},z^*)]{\mathrm{consensus}} \le (K-1)\exp\bigl(-m_{\mathrm{eff}}\bar{\delta}^{2}/8\bigr)$,
>
>    where $m_{\mathrm{eff}}$ counts only those views that are not deterministic functions of their predecessors (diversity condition) and still enjoy a positive average conditional margin $\bar{\delta} > 0$ (informativeness condition). This bound makes explicit that what drives exponential decay is *not* the raw number of generated views $m$, but the number of genuinely informative views $m_{\mathrm{eff}}$.
>
>    In the ideal independent case, $m_{\mathrm{eff}} = m$ and we recover the classical ensemble behaviour. In realistic correlated regimes, $m_{\mathrm{eff}}$ quickly saturates as new views become redundant, which explains the empirical plateau and clarifies how violations of independence affect the guarantees. The full derivation and interpretation are given in **Global Response**.
>
> 2. **Experimental scope and deep baselines.**
>    We have expanded the evaluation from two datasets to five standard text benchmarks (Reuters, DBpedia, AG News, Yahoo Answers, SearchSnippets) and compiled NMI/ARI/ACC results for 12 recent deep text clustering methods (2020–2025) alongside our consensus method. As discussed in **Global Response**, this shows that our pipeline, while simple, is competitive with more complex deep architectures, while our focus remains on understanding how *diverse informative views* improve strong single-view SBERT baselines.
>
> 3. **Computational cost, scalability, and hyperparameters.**
>    We now provide an explicit complexity analysis. The main overhead comes from the spectral consensus on the $N \times N$ co-occurrence matrix at training time, whereas inference reduces to a single GMM on the learned latent space with complexity comparable to standard single-view clustering. For reproducibility, we also add a concise hyperparameter table (including $\alpha, \beta, \tau$, number of views $m$, batch size, and learning rate) in the appendix. See **Global Response** for details.
>
> 4. **Consensus mechanism, co-occurrence matrix, and error plateau.**
>    We clarify that the co-occurrence matrix used for consensus is built by aggregating multi-view votes: each entry $(i,j)$ counts how many views assign documents $i$ and $j$ to the same cluster. In this sense, majority voting is implicitly used at the level of co-occurrence weights, and spectral clustering is then applied on this matrix to obtain the final partition.
>
> 5. **Figure clarity.**
>    We have enhanced the presentation of Figure 2 so that each metric is represented consistently and is easier to interpret.

---

### Author Response · Authors · 2025-11-25
**Global Response - Summary of revisions**

We thank all reviewers for their careful reading of our work and for their constructive feedback, comments, and suggestions, which we have carefully considered in revising the manuscript. The main changes concern (i) an extended theoretical analysis that explicitly handles correlated views by considering a martingale process and an *effective* number of views, (ii) a broader and better positioned empirical evaluation relative to recent works on deep clustering and multi-view consensus clustering, and (iii) a clarification of computational cost and scalability.

Our revisions address the following key aspects:

1. View dependence and theoretical assumptions.
2. Experimental scope and previous works.
3. Computational cost and scalability.
4. Summary of contributions.

---

### Author Response · Authors · 2025-11-25
**Global Response - View dependence and theoretical assumptions**

**Proof 2-C:**

Fix a point whose true class is $a$, and a rival $b \neq a$. For each view $v \in$ \{1,...,m\}, let
$X_v = \mathbf{1}{\text{ view } v \text{ assigns the point to } a}$ and
$Y_v = \mathbf{1}{\text{ view } v \text{ assigns the point to } b}$,
and define:

$Z_v = X_v - Y_v \in$ \{-1, 0, +1\}


$S = \sum_{v=1}^m Z_v$,
$X = \frac{1}{m}\sum_{v=1}^m X_v$,
$Y = \frac{1}{m}\sum_{v=1}^m Y_v$,
and $Z = X - Y$.

Misclassification of a point from class $a$ in favour of $b$ implies $S \le 0$.

We now allow the views to be correlated across $v$. Let
$F_t = \sigma(Z_1,\dots,Z_t)$ for $t=0,\dots,m$ be the natural filtration generated by the votes, and we consider the process $(M_t)$ to be a martingale with respect to the filtration $(F_t)$ for $t = 0,\dots,m$.

**Effective number of views:**

Some later views can be deterministic functions of the previous ones. Say that a view $v$ is degenerate if there exists a measurable function $f_v$ such that
$Z_v = f_v(Z_1,\dots,Z_{v-1})$, so that
$\mathbb{E}[Z_v \mid F_{v-1},a] = Z_v$ and hence:

$D_v = Z_v - \mathbb{E}[Z_v \mid F_{v-1},a] = 0$

Such a view is perfectly dependent on its predecessors and does not contribute any new martingale fluctuation.

Let $I_{\mathrm{eff}} \subseteq $ \{1,...,m\}, be the set of indices of non-degenerate (effective) views, and define
$m_{\mathrm{eff}} = |I_{\mathrm{eff}}|$.
For the effective views, assume there exist ${\delta_v}{\in I_{\mathrm{eff}}}$ such that:

$E[Z_v \mid F_{v-1}, a] \ge \delta_v$ for all $v \in I_{\mathrm{eff}}$.

Then
$\sum_{v\in I_{\mathrm{eff}}} \mathbb{E}[Z_v \mid F_{v-1},a] \ge \sum_{v\in I_{\mathrm{eff}}} \delta_v$.
Define the average conditional margin over effective views
$\bar{\delta} = \frac{1}{m_{\mathrm{eff}}}\sum_{v\in I_{\mathrm{eff}}} \delta_v$,
so the total conditional advantage along the effective views is $m_{\mathrm{eff}}\bar{\delta}$ with $\bar{\delta}>0$ (condition).

**Martingale decomposition and Azuma–Hoeffding:**

For $v \in$ \{1,...,m\} define
$D_v = Z_v - \mathbb{E}[Z_v \mid F_{v-1},a]$ and
$M_t = \sum_{v=1}^t D_v$ with $M_0 = 0$.

Then $\mathbb{E}[D_v \mid F_{v-1},a]=0$, so $(M_t, F_t)$ is a martingale.

Since $Z_v \in$ \{-1, 0, +1\} and $\mathbb{E}[Z_v\mid F_{v-1},a]\in[-1,1]$, we have
$|D_v| = |Z_v - \mathbb{E}[Z_v \mid F_{v-1},a]| \le 2$ for all $v$,
and $D_v=0$ whenever $v\notin I_{\mathrm{eff}}$.

By construction:
$S = \sum_{v=1}^m Z_v
= \sum_{v=1}^m \mathbb{E}[Z_v \mid F_{v-1},a] + \sum_{v=1}^m D_v
= \sum_{v=1}^m \mathbb{E}[Z_v \mid F_{v-1},a] + M_m$.

Restricting to effective indices:
$\sum_{v=1}^m \mathbb{E}[Z_v \mid F_{v-1},a]
\ge \sum_{v\in I_{\mathrm{eff}}} \mathbb{E}[Z_v \mid F_{v-1},a]
\ge m_{\mathrm{eff}}\bar{\delta}$,
so
$S \ge m_{\mathrm{eff}}\bar{\delta} + M_m$.
Hence
${S \le 0} \subseteq {m_{\mathrm{eff}}\bar{\delta} + M_m \le 0} = {M_m \le -m_{\mathrm{eff}}\bar{\delta}}$,
and therefore
$P(S \le 0 \mid a) \le P(M_m \le -m_{\mathrm{eff}}\bar{\delta} \mid a)$.

Applying Azuma–Hoeffding to $(M_t)$ with increments $D_v$ and bounds $|M_t - M_{t-1}|\le c_v$ gives, for any $x>0$,
$P(M_m - M_0 \le -x) \le \exp\bigl(-x^2 / (2\sum_{v=1}^m c_v^2)\bigr)$.

Here $M_0=0$, $x=m_{\mathrm{eff}}\bar{\delta}$, and we take $c_v=2$ for $v\in I_{\mathrm{eff}}$ and $c_v=0$ otherwise, so
$\sum_{v=1}^m c_v^2 = \sum_{v\in I_{\mathrm{eff}}} 4 \le 4,m_{\mathrm{eff}}$,
and thus
$P(M_m \le -m_{\mathrm{eff}}\bar{\delta} \mid a)
\le \exp\bigl(-(m_{\mathrm{eff}}\bar{\delta})^2 / (8 m_{\mathrm{eff}})\bigr)
= \exp\bigl(-m_{\mathrm{eff}}\bar{\delta}^{2}/8\bigr)$.

For a fixed pair $(a,b)$:

$P(a\to b) = P(S \le 0 \mid a) \le \exp\bigl(-m_{\mathrm{eff}}\bar{\delta}^{2}/8\bigr)$,
and by a union bound over the $K-1$ rivals $b\neq a$:

$P(\text{point from class } a \text{ is misclassified})
\le (K-1)\exp\bigl(-m_{\mathrm{eff}}\bar{\delta}^{2}/8\bigr)$.

Thus the expected misclustering fraction under the consensus labelling satisfies:

$\mathbb{E}[h(\hat{z},z^*)]{\mathrm{consensus}}
\le (K-1)\exp\bigl(-m_{\mathrm{eff}}\bar{\delta}^{2}/8\bigr)$,
with $m_{\mathrm{eff}} = |I_{\mathrm{eff}}|$.

---

### Author Response · Authors · 2025-11-25
**Global Response - Interpretation and Theorems**

**Interpretation:**
The bound decays exponentially in the effective number of views
$m_{\mathrm{eff}}$ (the number of views that are not deterministic functions of their predecessors and that still enjoy a positive average conditional margin $\bar{\delta} > 0$). In the idealised regime where every new view is both informative and non-degenerate, we have $m_{\mathrm{eff}} = m$ and
$\mathbb{E}[h(\hat{z},z^*)]_{\text{consensus}} \le (K-1)\exp(- m \bar{\delta}^{2} / 8)$,
so the misclustering error vanishes exponentially fast as more views are added.

The picture changes as soon as correlations become strong. If a new view is essentially a function of the previous ones $(Z_v \approx f_v(Z_1,\dots,Z_{v-1}))$, then it contributes almost no new randomness ($D_v \approx 0$) and does not increase $m_{\mathrm{eff}}$. In the extreme case where only a finite subset of views are genuinely new, $m_{\mathrm{eff}}$ quickly saturates at a constant while the total number of generated views $m$ continues to grow. The exponent $m_{\mathrm{eff}}\bar{\delta}^2 / 8$ then also saturates, and the bound no longer forces the misclustering error to go to zero.

Our empirical behaviour fits precisely this scenario: the first generated views are sufficiently informative, so $m_{\mathrm{eff}}$ increases and the consensus error decreases. As more views are added, they become highly correlated with linear combinations of earlier ones, so $m_{\mathrm{eff}}$ stabilises and the error plateau emerges. In practice, this shows that what matters is not the raw number of views $m$, but the number of informative views $m_{\mathrm{eff}}$ that contribute non-trivially to the consensus.

**Theorems:**
For each view $v \in {1,\dots,m}$ and class $a \in {1,\dots,K}$,

let $r_a^{(v)}$ be the probability that a point from class $a$ is correctly assigned by view $v$, and $p_{ab}^{(v)}$ the probability that it is assigned to a rival class $b \neq a$.

Define the per-view advantage
$\delta_a^{(v)} = r_a^{(v)} - \max_{b\neq a} p_{ab}^{(v)}$.

Let $Z_v = X_v - Y_v \in$ \{-1,0,1\} denote the vote of view $v$, where $Z_v = 1$ assigns the point to cluster $a$, $Z_v = -1$ assigns it to the most challenging rival cluster $b$, and $Z_v = 0$ assigns it to another cluster.

**Theorem 1- (independent views – idealised case):**

Assume:

- **Informativeness:** there exists $\delta > 0$ such that, for all classes $a$ and all views $v$,
$\delta_a^{(v)} \ge \delta$, equivalently $\mathbb{E}[Z_v \mid a] \ge \delta$.

- **Diversity:** the variables $(Z_v)_{v=1}^m$ are mutually independent conditional on the true class $a$.

Then the expected misclustering rate of the consensus labelling satisfies:

$\mathbb{E}[h(\hat{z}, z^*)]_{\text{consensus}} \le (K-1)\exp(- m \delta^2 / 2)$, which tends to $0$ as $m \to \infty$.

In particular, for $m$ large enough, the consensus error is strictly smaller than the single-view error.

**Theorem 2 - (correlated views – realistic case):**

Let $F_t = \sigma(Z_1,\dots,Z_t)$ be the natural filtration generated by the votes, and suppose there exists a set $I_{\mathrm{eff}} \subseteq {1,\dots,m}$ of effective views with cardinality $m_{\mathrm{eff}} = |I_{\mathrm{eff}}|$ such that:

- **Informativeness:** for each $v \in I_{\mathrm{eff}}$ there exists $\delta_v$ with
$\mathbb{E}[Z_v \mid F_{v-1}, a] \ge \delta_v$,
and the average conditional margin is bounded below:

$\frac{1}{m_{\mathrm{eff}}}\sum_{v\in I_{\mathrm{eff}}} \delta_v \ge \bar{\delta} > 0$.

- **Diversity:** views outside $I_{\mathrm{eff}}$ are deterministic functions of their predecessors and hence do not contribute new randomness; only the $m_{\mathrm{eff}}$ effective views provide genuinely new information.

Then the expected misclustering rate of the consensus labelling satisfies:

$\mathbb{E}[h(\hat{z}, z^*)]{\text{consensus}} \le (K-1)\exp(- m_{\mathrm{eff}} \bar{\delta}^2 / 8)$.

In particular, the error decays exponentially in the effective number of views $m_{\mathrm{eff}}$. In the ideal regime where every new view is informative and non-degenerate, $m_{\mathrm{eff}} = m$ and we recover the behaviour of the independent-case theorem. When many views are highly correlated, $m_{\mathrm{eff}}$ quickly saturates and the bound no longer forces the misclustering error to vanish, which matches the empirical plateau observed when adding redundant views.

---

### Author Response · Authors · 2025-11-25
**Global Response - Experimental scope and baselines**

We broaden and better position our empirical study to clarify where our method sits relative to existing deep clustering work. First, we extend the evaluation from two datasets to five standard text benchmarks (Reuters, DBpedia, AG News, Yahoo Answers, SearchSnippets). Second, we systematically situate our results with respect to recent deep text clustering methods published between 2020 and 2025. Concretely, we collect and report NMI/ARI/ACC scores for 12 prior deep clustering models on these benchmarks (whenever available in the original papers) and present them in a new summary table in the revised manuscript, alongside the results of our consensus method. This shows that, despite its simple design, our approach is competitive with more complex architectures. At the same time, we emphasise that the primary goal of the paper is not to introduce yet another deep architecture, but to understand how generating diverse and informative views can systematically enhance single-view baselines. The simplicity of our construction makes the gains directly attributable to the multi-view consensus mechanism itself.

Table: Summary of deep text clustering works (2020–2025).
Each cell is NMI / ARI / ACC (in %).

| Paper (Year)                        | Reuters         | DBpedia        | AG News        | Yahoo Ans.     | SearchSnip.    |
|-------------------------------------|-----------------|----------------|----------------|----------------|----------------|
| Seed-Guided Deep Clust. (2020)      | -- / 61.1 / 81.4| -- / 67.0 / 80.3| -- / 64.4 / 84.8| -- / 34.4 / 61.1| -- / -- / --   |
| SCCL (2021)                         | --              | --             | 68.2 / -- / 88.2| --             | 71.1 / -- / 85.2|
| SCA-AE (2021)                       | --              | --             | 34.1 / -- / 68.3| 34.7 / -- / 56.0| 50.2 / -- / 68.7|
| BERT-Rep for Clust. (2022)          | 52.1 / 53.4 / 86.5| --          | 53.8 / 57.0 / 80.3| 30.1 / 24.7 / 48.7| --            |
| DFTC (2022)                         | 69.6 / 61.5 / 61.5| 80.2 / 79.2 / 76.0| 60.3 / 60.0 / 82.1| 51.1 / 35.0 / 34.4| --           |
| ADCluster (2023)                    | --              | --             | -- / -- / 83.44| -- / -- / 67.94| --              |
| TC-DWA (2023)                       | --              | 82.3 / 71.1 / 79.5| 56.6 / 60.4 / 82.7| --           | --              |
| Deep Emb. Clust. (2023)             | 71.5 / 69.1 / 79.1| --          | 64.2 / 67.2 / -- | 43.7 / 35.5 / 60.7| --             |
| TCBPMA (2024)                       | 55.4 / 61.4 / 76.0| --          | 58.6 / 64.4 / 84.4| 37.0 / 27.5 / 52.4| --             |
| HS-GC (2024)                        | --              | --             | 66.9 / -- / 87.7| --              | 68.2 / -- / 85.0|
| SDEC (2025)                         | 62.7 / 61.6 / 70.4| 80.8 / 78.7 / 75.5| 62.5 / 62.8 / 85.7| 36.3 / 34.8 / 53.6| --           |
| POTA (2025)                         | --              | --             | 66.5 / -- / 87.3| --              | 67.4 / -- / 80.1|
| **Our Consensus Clust.**            | 82.7 / 81.6 / 91.8| 81.7 / 71.9 / 81.0| 61.6 / 63.7 / 83.7| 49.7 / 45.8 / 66.2| 68.8 / 69.7 / 86.5|

---

### Author Response · Authors · 2025-11-25
**Global Response - Computational cost and scalability**

Let $N$ be the number of documents, $B$ the number of pre-trained encoders, $m$ the number of views, $K$ the number of clusters, $d$ the feature dimension, and $T$ the number of EM iterations.


1) BERT feature extraction has one-time cost
$O(B N)$, linear in $N$ and $B$.


2) For PCA, WPT, and Noise Injection, generating $m$ views scales as
$O(m N d)$, linear in the number of documents, views, and feature dimension.


3) For each view, GMM clustering via EM on $N$ points in $\mathbb{R}^d$ with $K$ components and $T$ iterations costs
$O(N K d T)$, hence $O(m N K d T)$ over $m$ views in total.


4) The spectral consensus step takes the $m$ per-view clusterings, builds an $N \times N$ co-occurrence matrix (cost $O(m N^2)$ in the worst case), and runs spectral clustering on it, with worst-case complexity $O(N^3)$ (or $O(N^2 K)$ with sparse/iterative eigensolvers).


Importantly, this spectral co-occurrence consensus is used only at training time to learn a rearranged latent space and align cluster identities across views.
At inference time we no longer recompute all $m$ views nor re-run spectral consensus: a single GMM on the learned latent representation yields cluster assignments with cost $O(N K d T)$, comparable to standard single-view clustering. Thus, the most expensive component is the spectral step with $O(N^3)$ complexity, but this cost is incurred once during training, while inference remains lightweight.

---

### Author Response · Authors · 2025-11-25
**Global Response - Summary of contributions**

**Summary of contributions.**
For clarity, we restate the main contributions of our work:

1. **Consensus clustering framework.**
   A consensus clustering framework that generates multiple views of the original embeddings using several transformations (SBERT, PCA, WPT, Gaussian noise, etc.) and aggregates these views via a spectral co-occurrence consensus step, yielding lower misclustering error than single-view baselines.

2. **Refined theoretical analysis.**
   A refined theoretical analysis that (a) recovers the classical exponential improvement in the ideal independent-view regime, and (b) extends it to correlated views by introducing the effective number of views $m_{\text{eff}}$, explaining the empirically observed performance saturation.

3. **Extensive empirical study.**
   An extensive empirical study on five standard benchmarks (Reuters, DBpedia, AG News, Yahoo Answers, SearchSnippets), with comparisons against 12 recent deep clustering methods (2020–2025), showing that our simple consensus + latent-space pipeline is competitive with more complex architectures.

4. **Generalization experiments.**
   Generalization experiments on unseen text, demonstrating that the learned latent space and consensus mechanism remain effective beyond the training corpora and that the method is easy to deploy for organising new, unseen documents.

---

> ### Comment · Reviewer_Az8T · 2025-11-26
> **Thank you for your response.**
>
> Thank you for your response. It clarified my questions and concerns. Let me update my evaluation score.

---

### Comment · Reviewer_bHgA · 2025-11-26
**Thank you for your detailed response**

The substantial expansion of theoretical analysis and experimental content inadvertently highlights the initial inadequacy of the submission.  More importantly, these additions still do not resolve the core issue of limited methodological innovation. Therefore, I maintain my initial rating.

---

### Note · Authors · 2025-12-03

**Comment:**

In light of the reviewers’ comments, there is a need to improve and clarify the work. We believe that doing so requires a careful and thorough revision, which will take additional time beyond the current review cycle.

Rather than asking the reviewers and area chairs to continue evaluating a version of the paper that we know will need revision, we prefer to withdraw the submission at this stage in order to save their time and effort.

We sincerely thank the reviewers and the program committee for their time and feedback, which will be very helpful for strengthening a future version of this work.

**Withdrawal Confirmation:**

I have read and agree with the venue's withdrawal policy on behalf of myself and my co-authors.